# 3-Hydroxyanthranilic Acid Delays Paralysis in *Caenorhabditis elegans* Models of Amyloid-Beta and Polyglutamine Proteotoxicity

**DOI:** 10.3390/biom14050599

**Published:** 2024-05-18

**Authors:** Bradford T. Hull, Kayla M. Miller, Caroline Corban, Grant Backer, Susan Sheehan, Ron Korstanje, George L. Sutphin

**Affiliations:** 1Molecular and Cellular Biology Department, University of Arizona, Tucson, AZ 85721, USA; 2Cancer Biology Graduate Interdisciplinary Program, University of Arizona, Tucson, AZ 85721, USA; 3The Jackson Laboratory, Bar Harbor, ME 04609, USA

**Keywords:** metabolism, tryptophan, kynurenine, Alzheimer’s disease, Huntington’s disease, neurodegeneration, proteotoxicity, *Caenorhabditis elegans*

## Abstract

Age is the primary risk factor for neurodegenerative diseases such as Alzheimer’s and Huntington’s disease. Alzheimer’s disease is the most common form of dementia and a leading cause of death in the elderly population of the United States. No effective treatments for these diseases currently exist. Identifying effective treatments for Alzheimer’s, Huntington’s, and other neurodegenerative diseases is a major current focus of national scientific resources, and there is a critical need for novel therapeutic strategies. Here, we investigate the potential for targeting the kynurenine pathway metabolite 3-hydroxyanthranilic acid (3HAA) using *Caenorhabditis elegans* expressing amyloid-beta or a polyglutamine peptide in body wall muscle, modeling the proteotoxicity in Alzheimer’s and Huntington’s disease, respectively. We show that knocking down the enzyme that degrades 3HAA, 3HAA dioxygenase (HAAO), delays the age-associated paralysis in both models. This effect on paralysis was independent of the protein aggregation in the polyglutamine model. We also show that the mechanism of protection against proteotoxicity from HAAO knockdown is mimicked by 3HAA supplementation, supporting elevated 3HAA as the mediating event linking HAAO knockdown to delayed paralysis. This work demonstrates the potential for 3HAA as a targeted therapeutic in neurodegenerative disease, though the mechanism is yet to be explored.

## 1. Introduction

Neurodegenerative diseases have become increasingly prevalent as our population has shifted to an older demographic over the last 30 years. Alzheimer’s disease (AD) is now the seventh leading cause of the death in the United States and the most common form of neurodegeneration in the elderly [1]. As our cells age, a functional decline in the machinery that maintains protein homeostasis results in accumulation of damaged, misfolded, and aggregate-prone proteins [2,3,4]. The disruption of proteostasis is considered a hallmark of aging [5,6]. The aggregation of specific protein species is a defining feature—and is thought to be involved in the pathogenesis—of age-associated neurodegenerative diseases including AD and Huntington’s disease (HD) [7,8,9,10,11,12]. Current treatments for these diseases generally target symptoms (cholinergic and glutamatergic signaling in neurons in AD; huntingtin production in HD), but do not address the underlying pathology [13,14,15]. Hundreds of clinical trials targeting different aspects of AD, HD, and other neurodegenerative disease neuropathology—amyloid toxicity, tau toxicity, huntingtin production, oxidative stress, and neuroinflammation—have largely failed to demonstrate efficacy, with some limited success in delaying symptomatic progression [16,17,18]. Identifying effective treatments for AD and HD is a major current focus of scientific resources, and there is a critical need for novel therapeutic strategies.

The kynurenine pathway (Figure 1) is a highly conserved metabolic pathway that converts ingested tryptophan to nicotinamide adenine dinucleotide (NAD+) [19]. Several studies have shown promise in delaying the aging and age-associated decline in protein homeostasis through the inhibition of tryptophan degradation via the kynurenine pathway. In the roundworm *Caenorhabditis elegans*, proteotoxicity is commonly modeled by expressing aggregate-prone proteins in body wall muscle, which causes the accumulation of muscle protein aggregates and accelerated paralysis with age [20,21]. van der Goot et al. [22] reported that knockdown of the gene *tdo-2*, encoding the kynurenine pathway enzyme tryptophan 2,3-dioxygenase (TDO), delayed the age-associated pathology in several of these models, including worms expressing human amyloid-beta (Aβ) or a 35-unit polyglutamine tract (Q35), modeling proteotoxicity in AD and HD, respectively. We validated this observation and further found that knockdown of *kynu-1* encoding the kynurenine pathway enzyme kynureninase (KYNU) similarly delays paralysis in these models [23]. Both interventions extend the lifespan in wild-type animals, supporting the broader link between the maintenance of protein homeostasis and aging [22,23]. We recently reported that knockdown of *haao-1*, a third kynurenine pathway gene encoding the enzyme 3-hydroxyanthranilate 3,4-dioxygenase (HAAO), increases the lifespan in *C. elegans* [24]. Worms lacking *haao-1* accumulate the HAAO metabolic substrate, 3-hydroxyanthranilic acid (3HAA), and supplementing 3HAA in the growth media similarly extends the lifespan [24]. 

Here, we examine the impact of *haao-1* inhibition on Aβ and Q35 proteotoxicity in *C. elegans*. We find that *haao-1* knockdown delays paralysis in both contexts but does not prevent Q35 protein aggregation. Supplementing 3HAA similarly delays paralysis. The effectiveness at delaying paralysis was also dependent on environmental temperature. We include previously published data on knockdown of *tdo-2* or *kynu-1* in these models for comparison. These results suggest that the inhibition of HAAO or supplementation with 3HAA may hold potential as a clinical intervention for neurodegenerative disease.

## 2. Materials and Methods

### 2.1. C. elegans Strains and Maintenance

The following strains were obtained from the *Caenorhabditis* Genetic Center (CGC) in the College of Biological Sciences at the University of Minnesota: *dvIs2[pCL12(unc-54p::Aβ_1-42_) + rol-6(su1006)]* (CL2006) and *rmIs132[unc-54p::Q35::YFP]* (AM140). We maintained worms on solid nematode growth media (NGM) seeded with *Escherichia coli* strain HT115 bacteria at 20 °C as previously described [25]. Worms were age-synchronized via bleach prep and transferred to plates containing 50 μM 5-fluorodeoxyuridine (FUdR) to prevent reproduction at the L4 larval stage [25].

### 2.2. RNA Interference

We conducted RNA interference (RNAi) feeding assays using standard media preparation procedures [25]. RNAi feeding bacteria were obtained from the Ahringer *C. elegans* RNAi feeding library [26,27]. All RNAi plasmids were sequenced to verify the correct target sequence. All RNAi experiments were conducted on NGM containing 1 mM Isopropyl β-D-1 thiogalactopyranoside (IPTG) to activate production of RNAi transcripts and 25 μg/mL carbenicillin to select RNAi plasmids and seeded with live *E. coli* (HT115) containing RNAi feeding plasmids. Worms were raised on egg plates seeded with RNAi bacteria until the L4 larval stage, and then transferred to FUdR-containing plates seeded with RNAi bacteria as described above. 

### 2.3. 3HAA Supplementation

3HAA supplementation was achieved by adding the appropriate mass of solid 97% pure 3HAA (Sigma-Aldrich, St. Louis, MO, USA; catalog number 148776) to NGM media during preparation prior to autoclaving. Worms were raised on egg plates containing 1 mM 3HAA until the L4 larval stage, and then transferred to FUdR-containing plates containing 1 mM 3HAA as described above.

### 2.4. Paralysis Analysis

Paralysis experiments were conducted as previously described [23,25]. Briefly, adult worms were maintained on NGM RNAi or 3HAA plates with FUdR for the duration of their lives. Animals were examined every 1–2 days (25 °C) or 2–3 days (15 °C) by nose- and tail-prodding with a platinum wire pick. Worms were scored as paralyzed if they were unable to move relative to the plate surface. Worms showing vulva rupture were included in all analyses and worms that fled the surface of the plate were excluded. *p*-values for statistical comparison of paralysis between test groups were calculated using the log-rank test via the *survdiff* function in the R (version 4.3.2) “survival” package (version 3.5-7). For paralysis experiments, raw data are provided in Appendix A and summary statistics are provided in Appendix A.

### 2.5. Polyglutamine Aggregate Imaging and Analysis

Polyglutamine aggregate imaging and quantification was completed as previously described [23]. Briefly, we quantified polyglutamine aggregate number and volume using 3D fluorescence microscopy of transgenic worms expressing fluorescently tagged polyglutamine repeats (Q35::YFP) at 7, 12, and 17 days of age at 15 °C, and 7 and 11 days of age at 25 °C. At least 8 animals were examined for each RNAi at each age. Worms were immobilized using 25 nM sodium azide in M9 buffer on microscope slides with 6% agarose pads. Worms were imaged using a Leica TCS SP8 Laser Scanning Microscope equipped with a 10× objective (Leica Microsystems Inc., Deerfield, IL, USA). Z-stack images were collected for whole worms with stack spacing selected based on the Nyquist sampling theorem (minimum 2.3 images per minimum aggregate diameter). The microscope was set to capture images with resonance scanning and line averaging of 16. Images for worms that did not fit a single field of view were merged with XuvStitch version 1.8.099x64. Protein aggregates in each worm were detected and analyzed using Imaris x64 version 8.1.2. The surface tool was used to identify aggregates with thresholds for minimum fluorescence intensity and aggregate volume to ensure the analyzed surfaces were aggregates and not discrete proteins: surface = 3.0, absolute intensity = 75, seed points = 5:15, intensity > 150, volume > 45, and number of voxels > 500. Aggregate volume and number were exported to R for analysis. Two-sided Welch’s *t*-tests were used to determine significance in observed differences between target and EV RNAi at each age. For aggregate quantification experiments, raw data are provided in Appendix A and summary statistics are provided in Appendix A.

## 3. Results

To test the efficacy of *haao-1* disruption at improving proteotoxicity linked to neurodegeneration, we used RNAi to knockdown *haao-1* in *C. elegans* transgenically expressing polyglutamine fused to a yellow fluorescence protein (Q35::YFP) or human Aβ in body wall muscle, modeling the proteotoxicity in HD and AD, respectively. We include data for both *tdo-2* and *kynu-1* knockdown for comparison [23]. Similar to previous reports for knockdown of *tdo-2* or *kynu-1* [22,23], *haao-1* knockdown delayed the age-associated paralysis in worms at 15 °C (Figure 1b,c). At 25 °C, the effect was present but much smaller in both models (Figure 1d,e). These results show that *haao-1* disruption may be protective against AD- and HD-associated proteotoxicity.

We next investigated whether the mechanism of delayed paralysis in Q35::YFP worms might be a consequence of the inhibition of Q35 protein aggregate formation, a hallmark of HD. We used confocal microscopy to quantify the number and size of Q35::YFP aggregates at different ages for worms cultured at both 15 °C and 25 °C. *haao-1* knockdown did not affect either the number or volume of protein aggregates at either 15 °C (Figure 2) or 25 °C (Figure 3), with the exception of a small but significant reduction in aggregate volume at 25 °C in 11-day-old worms (Figure 3b). This is in contrast to our previously reported observation that *tdo-2(RNAi)* can reduce the number of aggregates at 25 °C (Figure 3a), and aggregate volume at both 15 °C (Figure 2b) and 25 °C (Figure 3b), and that *kynu-1(RNAi)* can reduced the mid-life aggregate volume at 15 °C (Figure 2b) [23]. This suggests that the protective effects of *haao-1* knockdown against polyglutamine toxicity are independent of protein aggregation. 

In an earlier work, we found that the beneficial effects of *haao-1* knockdown on the lifespan are mediated by increased physiological levels of the HAAO substrate metabolite 3HAA (Figure 1). Animals lacking functional *haao-1* accumulate 3HAA with age, and a 1 mM 3HAA supplementation in the growth media is sufficient to extend the mean *C. elegans* lifespan by 22%, a similar degree as knockout (26.2%) or RNAi knockdown (21.9%) of *haao-1* [24]. Meek et al. [28] predicted, in silico, that 3HAA should bind human Aβ at a critical region for protein misfolding, and found, in vitro, that 3HAA prevented protein Aβ aggregation. To investigate whether 3HAA supplementation can similarly recapitulate the protective effects of *haao-1(RNAi)* against proteotoxicity in *C. elegans*, we supplemented worms expressing muscle Q35 and Aβ with 1 mM 3HAA. The 1 mM 3HAA supplementation delayed the age-associated paralysis to a nearly identical degree as experiment-matched worms subjected to *haao-1(RNAi)* in both Q35 (Figure 4a) and Aβ (Figure 4b) animals. This observation supports a model in which elevated physiological 3HAA mediates the protection against proteotoxicity by *haao-1* disruption.

## 4. Discussion

Here, we describe a small study examining the impact of *haao-1* disruption or 3HAA supplementation on the pathology in *C. elegans* models of AD- and HD-associated proteotoxicity. We first showed that *haao-1* knockdown delayed the age-associated paralysis in worms expressing Aβ or Q35::YFP, suggesting that *haao-1* disruption may be protective against proteotoxicity in AD and HD, respectively. We next investigated the mechanism of delayed paralysis and found that *haao-1* knockdown did not meaningfully affect either the number or volume of Q35::YFP protein aggregates. This suggests that the protective effect of *haao-1* knockdown against polyglutamine proteotoxicity in the *C. elegans* muscle is independent of protein aggregation. Because the transgenic Aβ construct is not labeled, we did not evaluate the Aβ aggregation, leaving open the possibility that 3HAA may improve proteotoxicity by preventing aggregation in this model. Finally, we showed that media supplementation with 1 mM 3HAA closely mimicked the benefit of *haao-1* knockdown on the age-associated paralysis, supporting a model in which the protection against proteotoxicity by *haao-1* disruption is mediated by elevated physiological 3HAA. 

These observations suggest that *haao-1* disruption likely acts to reduce proteotoxicity via a mechanism that is at least partially distinct from both *tdo-2* and *kynu-1* for two reasons. First, *haao-1* disruption appears to be acting by elevating physiological levels of 3HAA. 3HAA levels are not elevated by the disruption of either *tdo-2* or *kynu-1* as we and others previously reported [22,23,24]. Specifically, *haao-1* knockdown resulted in an age-dependent increase in 3HAA that was 40-fold or higher relative to wild-type ones by 14 days of age, while *tdo-2* or *kynu-1* knockdown resulted in 3HAA levels at or below wild-type ones [22,23,24]. In contrast, *tdo-2* knockdown results in elevated tryptophan (TRP) while *kynu-1* knockdown results in elevated kynurenine (KYN) and 3-hydroxykynurenine (3HK) [22,23,24], as suggested by the structure of the kynurenine pathway (Figure 1a). Second, *haao-1* disruption does not impact the Q35::YFP aggregate size or volume, while the disruption of both *tdo-2* and *kynu-1* affects the aggregate number or volume in at least one context (Figure 2 and Figure 3). This leaves open the possibility that these interventions act via both common mechanisms (such as reducing production downstream metabolites) and distinct mechanisms (with *haao-1* knockdown elevating 3HAA and *tdo-2* and *kynu-1* acting via a separate mechanism). The lifespan extension from the disruption of *tdo-2*, *kynu-1*, and *haao-1* was partially dependent on different combinations of established aging pathways [23,24]. Evidence presented by van der Goot et al. [22] suggests that the protection of *tdo-2* knockdown against pathology in a *C. elegans* model of alpha-synuclein proteotoxicity (modeling Parkinson’s disease) may be mediated by elevated physiological levels of tryptophan. These observations provide guidance for future detailed mechanistic studies of the broader mechanisms mediating the protection of kynurenine pathway interventions against proteotoxicity. 

The role of 3HAA in neurodegenerative disease has only been explored in a few small studies. Meek et al. [28] showed, in silico, that 3HAA was capable of binding to and preventing the misfolding of Aβ in an energetically favorable reaction. Other groups have found evidence linking 3HAA to an improved prognosis and survival from spinal cord injury [29,30] and the repression of pro-inflammatory signaling and behavior of innate immune cells [31,32,33,34,35,36], including both astrocytes and microglia [37]. Parrott et al. [38] found that HAAO knockout mice were protected against behavioral despair induced by lipopolysaccharide (LPS)-induced inflammation. Together, these studies suggest that elevating 3HAA may have specific benefits in preventing Aβ aggregation in AD and broader benefits in neurodegenerative disease by reducing neuroinflammation.

Quinolinic acid (QA), another metabolite of the kynurenine pathway which is created from 3HAA by HAAO (Figure 1a), has been described as an endogenous neurotoxin produced in active microglia and implicated in several human neurological diseases, including AD and HD [39,40,41]. Tau, an endogenous microtubule-associated protein whose phosphorylation and subsequent aggregation is considered a pathological hallmark of AD, has been shown to have elevated phosphorylation levels in the presence of QA [42]. QA has also been proposed to modulate excitotoxic neuronal death in HD [43]. Combined, this evidence suggests that HAAO disruption may be doubly beneficial for the treatment of neurodegenerative disease pathogenesis in both increasing 3HAA and decreasing QA. Reduced QA is also a candidate for a shared mechanism of action for the benefits of knocking down *tdo-2*, *kynu-1*, and *haao-1* in *C. elegans* Aβ and Q35 proteotoxicity models (Figure 1b–e) [23]. While the pathway structure suggests that QA should be reduced by all three interventions (Figure 1a), we have found QA challenging to measure via mass spectrometry relative to other metabolites in the kynurenine pathway [23,24] and have not yet been successful in developing a consistent protocol for QA quantification. 

*C. elegans* are poikilothermic and temperature is a major determinant of lifespan, with worms cultured at 15 °C living almost twice as long on average as worms cultured at 25 °C [44,45]. In this work, we initially examined each intervention at 15 °C and 25 °C, temperatures near the extremes of the viable range (~12 to 26 °C). The expression of both Aβ in strain CL2006 [46] and Q35::YFP in strain AM140 [21] are driven by the constitutive muscle-specific *unc-54* promoter which is not known to be temperature-sensitive, and we observe age-dependent paralysis in untreated animals across the temperature spectrum (Figure 1b–e and Figure 2a,b). In an earlier work, we found that the lifespan extension from *tdo-2*, but not *kynu-1* or *haao-1*, was dependent on environmental temperature [24]. Surprisingly, all three interventions were more effective at delaying paralysis in both Aβ and Q35 animals at 15 °C relative to 25 °C (Figure 1b–e). Interestingly, the benefits of *haao-1* knockdown were similar for Q35, but reduced for Aβ (Figure 3a,b), at 20 °C relative to 15 °C (Figure 1b,c), suggesting that these models may have a differential sensitivity to environmental temperature. Unlike *tdo-2(RNAi)* and *kynu-1(RNAi)*, which reduced both the number and volume of protein aggregates, particularly at higher temperatures, *haao-1* knockdown did not significantly alter protein aggregation, except for a slight reduction in aggregate volume in 11-day-old worms at 25 °C (Figure 3b). This differential effect suggests that the protective mechanisms of *haao-1* disruption against polyglutamine toxicity are independent of the alterations in protein aggregation, highlighting the complexity of the kynurenine pathway in modulating proteotoxicity in neurodegenerative diseases. The temperature-dependent differences observed in our study shed light on the nuanced effects of *haao-1* disruption on proteotoxicity associated with neurodegenerative diseases in *C. elegans* models. The implications for this temperature dependence for proteotoxicity in mammals are unclear, as, unlike *C. elegans*, human tissues maintain their temperature within a narrow range. However, because interventions that influence the *C. elegans* lifespan are often differentially effective at different temperatures [44], pathways that modulate the lifespan at 15 °C may be worth examining as potential mediators of 3HAA-dependent improvements in proteotoxicity.

Related to the environmental temperature, our practice is to not remove animals with age-associated vulva integrity defects (Avid [45], aka vulva rupture, as noted in Materials and Methods), which may alter interpretation. While many studies make a practice of removing Avid worms from analysis, many do not and there is not a clear standard in the field. A detailed examination in previous work suggests that Avid is a form of age-associated pathology in *C. elegans* that is influenced by pathways that also impact the lifespan [45]. As a pathology of aging, we cannot see a clear justification from excluding these animals from analysis. We acknowledge that the rupture may be confounded with paralysis if both are influenced by the intervention under study. However, removing Avid worms does not fix this issue, but simply shifts the bias in a different direction; removing ruptured worms may preferentially remove worms that will experience earlier or later paralysis, depending on the relationship between the intervention, Avid, and paralysis. Based on our experience, paralysis in the Aβ and Q35 models tends to occur prior to a significant onset of Avid in most cases (Figure 1 and Figure 3) [45], so we believe that the potential impact of choosing to either remove or retail ruptured animals in the analysis in this case is minimal. 

Finally, we note that, while our results are positive, there are inherent limitations with these *C. elegans* proteotoxicity models. First, these models express Q35 or Aβ in body wall muscle instead of neurons, a distinct cellular context from the proteotoxicity that occurs in neurodegenerative disease. Second, while *C. elegans* do have cells that carry out analogous roles to the mammalian innate immune system and share many conserved immune-signaling pathways with mammals, they lack many elements of mammalian immunity, including an analogous adaptive immune system. This may be relevant to the present study because the kynurenine pathway is active in immune cells and responsive to inflammatory signaling, and kynurenine metabolites produced by glial cells have been implicated in neurodegenerative pathology [42,43]. Third, the Q35 polyglutamine repeat model of HD disease mimics the polyglutamine expansion in the huntingtin gene that causes human HD but does not contain the full-length huntingtin gene. These considerations limit a strong interpretation of the presented data with respect to the impact of 3HAA and HAAO in human AD and HD. However, this study and the related literature discussed above motivates future detailed mechanistic studies of 3HAA supplementation and HAAO inhibition in worm models of neuronal Aβ and polyglutamine, which have a more subtle pathology than the muscle transgenic strains used in this study, and in preclinical rodent models.

## 5. Conclusions

In this study, we show that HAAO disruption improves the pathology in *C. elegans* models of AD- and HD-associated proteotoxicity. Additionally, we show that 3HAA supplementation is sufficient to mimic the effects of *haao-1* knockdown, suggesting that these beneficial effects are a result of 3HAA accumulation, consistent with our prior work demonstrating that the lifespan benefits of *haao-1* knockdown are also mediated by 3HAA in wild-type animals [24]. While our findings provide valuable insights into potential mechanisms underlying proteotoxicity and therapeutic interventions, *C. elegans* lack physiological features relevant to the pathophysiology of both AD and HD. Even considering these limitations, our study highlights 3HAA as a potential therapeutic target for both AD and HD. Further mechanistic studies are warranted to elucidate the precise pathways through which *haao-1* disruption and 3HAA supplementation exert their protective effects. Additionally, investigating these interventions in more clinically relevant models, such as neuronal Aβ and polyglutamine *C. elegans* models and preclinical rodent models, will be important steps for validating the therapeutic potential of HAAO and 3HAA.

## Figures and Tables

**Figure 1 biomolecules-14-00599-f001:**
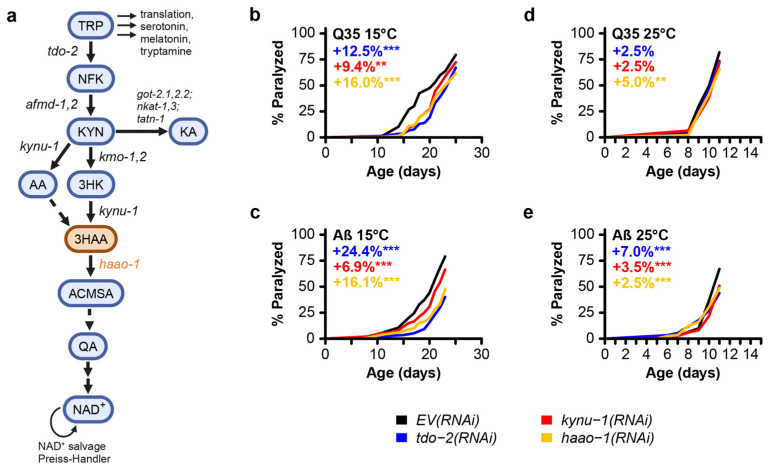
Knockdown of the kynurenine pathway gene *haao-1* delays paralysis in *C. elegans* models of polyglutamine and amyloid-beta proteotoxicity. (**a**) Schematic of the kynurenine pathway in *C. elegans*. Knockdown of *haao-1* substantially delayed age-associated pathology in *C. elegans* expressing either (**b**) Aβ or (**c**) a 35-unit polyglutamine tract (Q35) in body wall muscle at 15 °C to a similar degree as previously reported knockdown of *tdo-2* or *kynu-1* [23]. Knockdown of *haao-1* slightly delayed age-associated pathology in *C. elegans* expressing either (**d**) Aβ or (**e**) a Q35 in body wall muscle at 25 °C to a similar degree as previously reported knockdown of *tdo-2* or *kynu-1* [23]. Colored numbers indicate the percent change in mean age of paralysis vs. *EV(RNAi)*. ** *p* < 0.01, *** *p* < 0.001 for log-rank test vs. *EV(RNAi)*.

**Figure 2 biomolecules-14-00599-f002:**
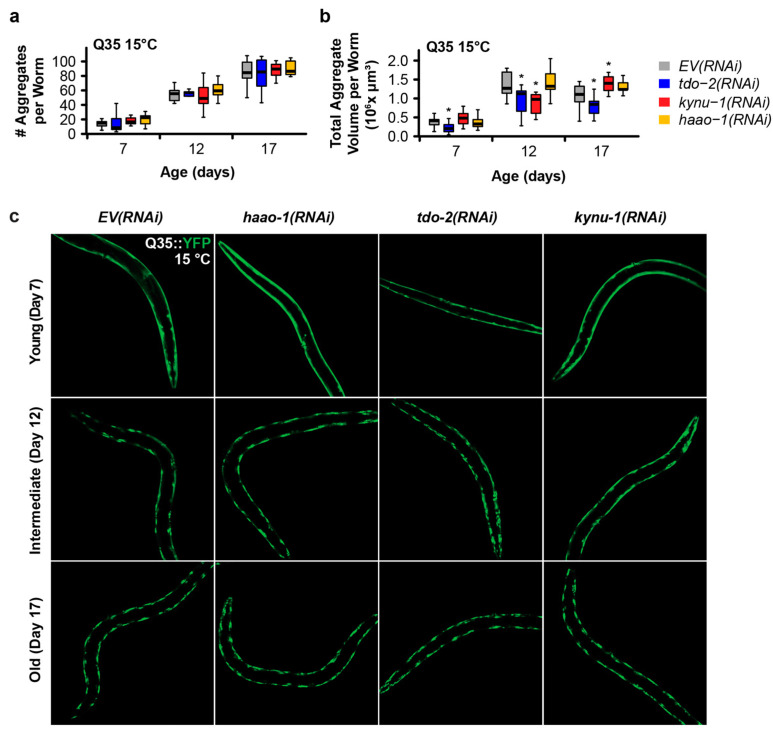
Knockdown of *haao-1* does not prevent polyglutamine aggregation at 15 °C. (**a**) *haao-1* knockdown did not affect the number of YFP-labeled muscle polyglutamine (Q35::YFP) protein aggregates at 15 °C, similar to our previous report for knockdown of *tdo-2* or *kynu-1* [23]. (**b**) *haao-1* knockdown did not affect the volume of Q35::YFP protein aggregates at 15 °C, unlike our previous report for knockdown of *tdo-2*, which reduced Q35::YFP aggregate volume at all ages examined, or *kynu-1*, which reduced Q35::YFP volume in mid-life [23]. (**c**) Representative images of Q35::YFP aggregation in each test group and age at 15 °C. * *p* < 0.05 for two-sided Welch’s *t*-test vs. age-matched *EV(RNAi)*.

**Figure 3 biomolecules-14-00599-f003:**
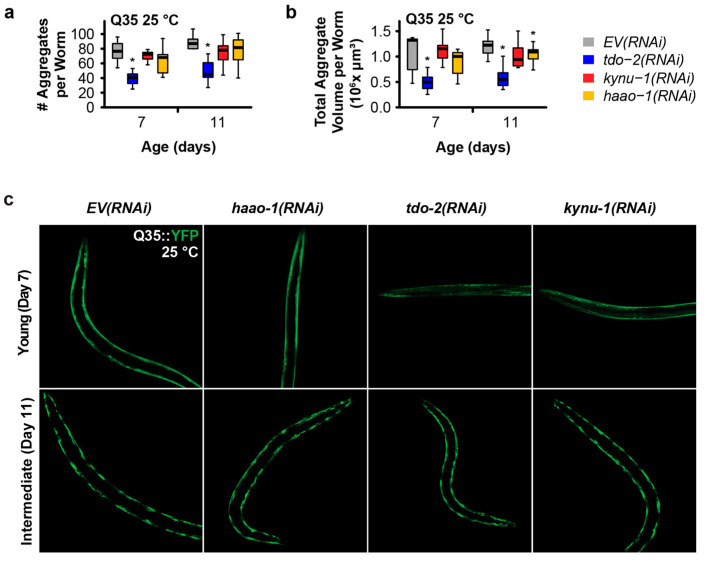
Knockdown of *haao-1* does not prevent polyglutamine aggregation at 25 °C. (**a**) *haao-1* knockdown did not affect the number of YFP-labeled muscle polyglutamine (Q35::YFP) protein aggregates at 25 °C, similar to our previous report for knockdown of *kynu-1* and unlike knockdown of *tdo-2*, which reduced Q35::YFP aggregate number at all ages examined [23]. (**b**) *haao-1* knockdown slightly reduced the volume of Q35::YFP protein aggregates at 25 °C, unlike our previous report for knockdown of *tdo-2*, which substantially reduced Q35::YFP aggregate volume both at all ages examined, or *kynu-1*, which did not reduce Q35::YFP volume [23]. (**c**) Representative images of Q35::YFP aggregation in each test group and age at 25 °C. * *p* < 0.05 for two-sided Welch’s *t*-test vs. age-matched *EV(RNAi)*.

**Figure 4 biomolecules-14-00599-f004:**
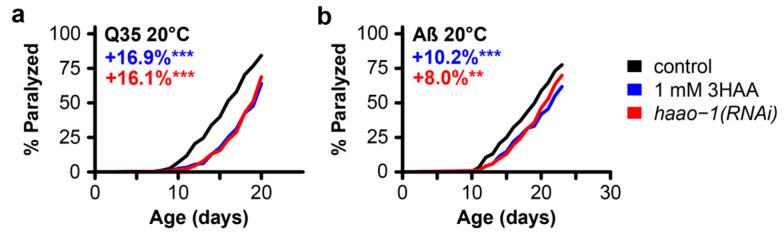
3HAA supplementation recapitulates the protective effects of *haao-1* knockdown against polyglutamine and Aβ proteotoxicity in *C. elegans*. Supplementing the growth media with 1 mM 3HAA in both Huntington’s (**a**) and Alzheimer’s (**b**) models, supplementation with 3HAA closely mimicked the effects of haao-1 knockdown. Colored numbers indicate the percent change in mean age of paralysis vs. control. ** *p* < 0.01, *** *p* < 0.001 for log-rank test vs. *EV(RNAi)*.

## Data Availability

The raw data for paralysis and aggregate quantification are provided in Appendix A. The raw images for Q35::YFP aggregation are available for download at: https://omero.jax.org/webclient/?show=project-1601 (accessed on 15 April 2024).

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
