# Peer review of "3-Hydroxyanthranilic Acid Delays Paralysis in Caenorhabditis elegans Models of Amyloid-Beta and Polyglutamine Proteotoxicity"

_biomolecules, 2024, doi:10.3390/biom14050599_

Round 1
Reviewer 1 Report
Comments and Suggestions for Authors
In this paper Hull et al. examine the effects of knocking down 3-hydroxyanthanilic acid (3HAA) dioxygenase (haao-1) in C. elegans models expressing either AB or expanded polyglutamine in body wall muscle (models of Alzheimer’s and Huntington’s disease). They find that haao-1 RNAi decreases paralysis in both models but does not affect aggregation in the polyglutamine model suggesting that the mechanism is independent of aggregation. Finally, they show that treating directly with 3HAA also decreases paralysis in these models. Overall, the paper is clearly written and the data support the conclusions. However, it does not provide much insight into underlying mechanisms.
Major points
The authors propose that tdo-2 disruption, kynu-1 disruption and haao-1 disruption may be acting through different mechanisms, and propose roles for 3HAA (beneficial) and QA (detrimental). It would be interesting to measure the levels of 3HAA and QA when each of these three genes is disrupted either with RNAi or a genetic deletion. This would provide some mechanistic insight into potential differences between disrupting these different genes in the tryptophan degradation pathway.
Minor points
Line 41-43 Add references
Line 75 Check strain nomenclature
Line 102 Why were ruptured worms included in analysis? Normally these are excluded as the worms internal organs are directly exposed the environment and these worms inevitably have a shortened lifespan.
Why were experiments done at multiple temperatures? Why 15 degree and 25 degrees for Figures 1-3 and 20 degrees for Figure 4?
Line 186 It would be helpful for readers to mention the effect of 1 mM 3HAA on lifespan here so they don’t have to look it up
Line 214-215 This sentence was hard to understand. Please rephrase for clarity.
Author Response
Response to Reviewer 1:
In this paper Hull et al. examine the effects of knocking down 3-hydroxyanthanilic acid (3HAA) dioxygenase (haao-1) in C. elegans models expressing either AB or expanded polyglutamine in body wall muscle (models of Alzheimer’s and Huntington’s disease). They find that haao-1 RNAi decreases paralysis in both models but does not affect aggregation in the polyglutamine model suggesting that the mechanism is independent of aggregation. Finally, they show that treating directly with 3HAA also decreases paralysis in these models. Overall, the paper is clearly written and the data support the conclusions. However, it does not provide much insight into underlying mechanisms.
The reviewers make an important point about this work, which is indeed primarily reports the observation that 3HAA can delay paralysis in models of amyloid-beta and polyglutamine toxicity and largely does not address mechanism. At present, our focus is on understanding the impact of 3HAA on aging more broadly and we are not funded to pursue this line of inquiry related to neurodegenerative disease. Our hope is to do so in the future if we are able to obtain funding but believe that the reported observations will be of interest to the research community in the meantime and should not be unduly delayed.
Major points
The authors propose that tdo-2 disruption, kynu-1 disruption and haao-1 disruption may be acting through different mechanisms, and propose roles for 3HAA (beneficial) and QA (detrimental). It would be interesting to measure the levels of 3HAA and QA when each of these three genes is disrupted either with RNAi or a genetic deletion. This would provide some mechanistic insight into potential differences between disrupting these different genes in the tryptophan degradation pathway.
This is an important point. Both we and van der Goot et al. [PMID: 22927396] measured 3HAA and several other kynurenine metabolites in response to knockdown of tdo-2, kynu-1, and haao‑1. These animals have elevated TRP, KYN/3HK, and 3HAA, as the pathway structure suggests. Unfortunately, we have found QA very difficult to accurately quantify via mass spectrometry and have yet to develop a consistent protocol. Van der Goot et al. similarly did not report QA levels, and we suspect similar technical difficulties. This leave open the possibility for a common mechanism linking the three interventions through lowered QA production (which is also suggested by pathway structure), since QA is a known neuroexcitatory toxin, and we hope to solve the technical problem and test this idea in the future. We have added text to the discussion describing what has been reported in previous studies with respect to other kynurenine metabolites in additional detail.
Minor points
Line 41-43 Add references
Added.
Line 75 Check strain nomenclature
Corrected.
Line 102 Why were ruptured worms included in analysis? Normally these are excluded as the worms internal organs are directly exposed the environment and these worms inevitably have a shortened lifespan.
While many studies make a practice of removing ruptured worms from analysis, many do not and there is not a clear standard in the field. We have examined this phenotype in some detail in previous work [PMID: 27566309]. Our examination suggests that vulva rupture is a form of age-associated pathology in C. elegans that is influenced by pathways that also influence lifespan. As a pathology of aging, we cannot see a clear justification from excluding these animals from analysis. We acknowledge that rupture may be confounded with paralysis if both are influenced by the intervention under study. However, removing ruptured worms does not fix this issue, simply shifts the bias in a different direction (because removing ruptured worms may preferentially remove worms that will experience earlier or later paralysis, depending on the relationship between intervention, rupture, and paralysis). Based on our experience observing ruptured worms [PMID: 27566309], paralysis in the Aβ and Q35 models tends to occur before significant onset of rupture in most cases, so we believe that the potential impact in this case is minimal. At this stage, the best we can do is report what we did accurately so that the inclusion or rupture can be considered in interpretation. We now include a paragraph in the discussion outlining these points with appropriate references.
Why were experiments done at multiple temperatures? Why 15 degree and 25 degrees for Figures 1-3 and 20 degrees for Figure 4?
We should have made this more clear in our original submission. Temperature is a major determinant of C. elegans lifespan and a topic of several of our earlier studies. We previously found that the impact of tdo‑2 on C. elegans lifespan was temperature dependent, while the impact of haao-1 and kynu‑1 was not. We decided to include a temperature range in this study to see if the same was true for proteotoxicity-related paralysis. We added a paragraph discussing the impact of temperature on these phenotypes in the discussion.
Line 186 It would be helpful for readers to mention the effect of 1 mM 3HAA on lifespan here so they don’t have to look it up
The percent lifespan extension for 1 mM 3HAA, haao-1 knockout, and haao-1 RNAi knockdown are now included in the text for reference.
Line 214-215 This sentence was hard to understand. Please rephrase for clarity.
Rephrased for clarity.

Reviewer 2 Report
Comments and Suggestions for Authors
The article is well-structured, presented and written.
Minor comments:
-line 249: "Second, C. elegans while C. elegans".... it is repetition.
Comment on research:
-Has the level of the upstream and downstream metabolites measured? What are the evidence that the haao derived metabolites are gone?
- It is not clear (for a person not working in C.eleganc) why there is a temperature effect in the protection (15 vs 25)... Is it physiological? Is the expression of the constructs influenced by it? An explanation in the discussion should be added
- While the article is centred on Abeta and HD, the majority of labelling experiments have been done for HD. How can we assume that the same results would be observed in the AB context?
Author Response
Response to Reviewer 2:
The article is well-structured, presented and written.
Thank you, we hope the resubmission is similarly well presented.
Minor comments:
-line 249: "Second, C. elegans while C. elegans".... it is repetition.
This was a typo. Thanks for the catch. Corrected.
Comment on research:
-Has the level of the upstream and downstream metabolites measured? What are the evidence that the haao derived metabolites are gone?
A similar point was raised by the first reviewer. Please see details above.
- It is not clear (for a person not working in C. elegans) why there is a temperature effect in the protection (15 vs 25)... Is it physiological? Is the expression of the constructs influenced by it? An explanation in the discussion should be added.
Expression of both Aβ in strain CL2006 and Q35::YFP in strain AM140 are driven by the constitutive muscle-specific unc-54 promoter which is not known to be temperature sensitive, and we observe accelerated paralysis in untreated animals (relative to the standard timing for paralysis in wild type worms) across the temperature spectrum. A paragraph discussing temperature in this study is now included in the discussion.
- While the article is centered on Abeta and HD, the majority of labelling experiments have been done for HD. How can we assume that the same results would be observed in the AB context?
We cannot and the point is well received. Since the Aβ construct is not fluorescently labeled, we only examined aggregation in the Q35 model. We clarified the language in the discussion to reflect the possibility that 3HAA may improve Aβ toxicity by preventing aggregation. This is likely worth examining in the future.
